# Effect of Dietary or Supplemental Vitamin C Intake on Vitamin C Levels in Patients with and without Cardiovascular Disease: A Systematic Review

**DOI:** 10.3390/nu13072330

**Published:** 2021-07-08

**Authors:** Bianca J. Collins, Mitali S. Mukherjee, Michelle D. Miller, Christopher L. Delaney

**Affiliations:** 1Department of Nutrition and Dietetics, College of Nursing and Health Sciences, Flinders University, Bedford Park, SA 5042, Australia; bianca.collins@flinders.edu.au; 2Caring Futures Institute, College of Nursing and Health Sciences, Flinders University, Bedford Park, SA 5042, Australia; mitali.mukherjee@flinders.edu.au; 3Department of Vascular Surgery, Flinders Medical Centre, Flinders University, Bedford Park, SA 5042, Australia; chris.delaney@sa.gov.au

**Keywords:** vitamin C, supplementation, cardiovascular disease, plasma vitamin C

## Abstract

Atherosclerosis is a pro-oxidative and pro-inflammatory disease state, which is the underlying cause of most cardiovascular events, estimated to affect 5.2% of the Australian population. Diet, and specifically vitamin C, through its antioxidant properties can play a role in impeding the development and progression of atherosclerosis. This systematic review conducted comprehensive searches in Medline, Emcare, Scopus, PubMed, and Cochrane using key search terms for vitamin C, plasma vitamin C, supplementation, and cardiovascular disease (CVD). The results demonstrated that vitamin C supplementation resulted in a significant increase in vitamin C levels in populations with or without CVD, except for one study on the CVD population. It was also seen that the healthy population baseline and post-intervention vitamin C levels were high compared to the CVD population. However, further research is indicated for CVD population groups with varying baseline vitamin C levels, such as low baseline vitamin C, within a more representative elderly cohort in order to formulate and update vitamin C repletion guidelines.

## 1. Introduction

In 2014–2015, cardiovascular disease (CVD) was estimated to affect 1.2 million Australians (5.2%) [1]. CVD encompasses a group of diseases affecting the circulatory system, including coronary artery disease (CAD), stroke, heart failure, and peripheral artery disease (PAD) [1]. In 2018, CAD was the leading cause of death in Australia [2].

The underlying cause of most CVD events is atherosclerosis [3]. Atherosclerosis is a pro-inflammatory and pro-oxidative state which is preceded by endothelial cell dysfunction (ECD) [3,4,5]. Endothelial cells lining the cardiovascular system help maintain vascular homeostasis through various mechanisms, including the production of nitric oxide (NO), which acts as a vasodilator, decreases platelet aggregation, and blocks the expression of pro-inflammatory and adhesion molecules [6,7]. ECD is initiated by oxidative stress and is characterized by changes in the production and bioavailability of NO, resulting in an increased display of adhesion molecules on endothelial cells [8,9]. These changes cause the increased adhesion of platelets, T-cells, and monocytes, which can enter the intima along with low density lipoproteins (LDL), which can then become oxidized [8,9]. This initiates a perpetuating pathological sequence which results in the formation of an atherosclerotic plaque [9]. The destabilization of the plaque can result in life threatening cardiovascular events such as a stroke or a heart attack [3].

Diet can be used in the primary and secondary prevention of CVD by modification of risk factors through the antioxidant and anti-inflammatory properties of the diet [10,11,12]. Dietary patterns such as the Mediterranean diet, a dietary approach to stop hypertension (DASH), and generally healthy eating patterns all promote a diet high in fruit, vegetables, wholegrains, and legumes, and moderate amounts of fish, nuts, seeds, low fat dairy and red meat and poultry [10,11,12]. More specifically, vitamin C has been investigated to assess its effect on preventing the development and progression of atherosclerosis through its antioxidant properties [5]. Vitamin C can neutralize reactive oxygen species (ROS) which deplete tetrahydrobiopterin, a cofactor required for endothelial cell nitric oxide synthase (eNOS) [5]. Therefore, vitamin C can enhance eNOS, thus limiting pro-inflammation, adhesion, and platelet aggregation associated with atherosclerosis progression [7]. The interaction of vitamin C with vitamin E can aid in restoring the antioxidant function of oxidized vitamin E and help in impeding the atherogenic oxidative modification of LDL [5].

Vitamin C, also known as ascorbate, is a hydrophilic vitamin obtained through diet [5,7]. Oral intake results in its uptake through the intestine before its distribution around the body through the blood [13]. Vitamin C levels can be measured by a blood test, including serum or plasma vitamin C [14,15]. While there are differences in reference ranges of serum Vitamin C levels between different populations, one suggested reference range is 0.4–1.5 mg/dL or 23–85 µmol/L, which has been utilized for the purpose of this review [16,17]

Recent systematic reviews and meta-analyses of prospective observational studies found significant inverse associations between dietary and circulating concentrations of vitamin C and the risk of CVD and total CVD mortality [18,19]. Furthermore, findings suggest that vitamin C levels are more strongly associated with the risk of CVD and CVD mortality as compared to vitamin C dietary intake [18,19]. Conversely, an umbrella review of systematic reviews and meta-analyses found that large, randomized control trials (RCTs) reported no effect of vitamin C supplementation on major CVD outcomes [20]. Alternatively, a meta-analysis included in the umbrella review showed a statistically significant effect of vitamin C supplementation on endothelial cell function, with larger effects seen in subgroups with atherosclerosis [20]. There have been many suggested mechanisms for the contradictory findings between observational studies and RCTs, including inadequate amounts of vitamins supplemented in intervention trials [5].

A recent study found that 78% of patients with PAD were deficient in vitamin C [21]. The lower levels of vitamin C in patients with CVD has been attributed to a combination of factors including the increased turnover of vitamin C due to oxidation and inflammation associated with the disease process, and also a lower dietary intake of vitamin C [13]. However, a cross-sectional study observed that healthy and PAD participants had similar vitamin C intakes at 121 and 118 mg/day, respectively [22]. Despite the comparable intake of vitamin C, it was observed that serum vitamin C concentrations were significantly lower in the PAD group as compared to the healthy group (*p* = 0.0001) [22]. These findings would be consistent with a recent paper that suggested higher daily intake of vitamin C might be needed to sustain adequate levels of vitamin C in patients with diseases that cause increased vitamin C turnover, as compared to a healthy population [13].

Considering these factors, the question of whether vitamin C levels in the body adequately increase in response to vitamin C supplementation or intake in the CVD population is raised. Therefore, the aim of this review will be to determine if increasing dietary or supplemental intake of vitamin C will increase vitamin C levels in patients with and without CVD.

## 2. Methods

This systematic review was registered on the PROSPERO database and followed the PRISMA reporting guidelines.

### 2.1. Literature Search

A comprehensive search of Medline, Emcare, Scopus, ProQuest, and Cochrane databases was conducted on 10 August 2020. The search was executed using key words, or subject headings where applicable, including variations of vitamin C, dietary intake, supplementation, vitamin C concentration, and CVD. The full details of the search strategy can be found in Table 1. No restrictions were applied for publication date or language. If an article could not be sourced in English, Google Translate was used. No non-English papers were utilized in this review. Additionally, reference lists of relevant review articles were manually searched.

### 2.2. Study Selection

The inclusion and exclusion criteria set for the review have been depicted in Table 2. For studies to be included in the review, they needed to meet all the inclusion criteria outlined in Table 2. Search results were exported to EndNote X9 and subsequently uploaded to Covidence where duplicates were removed. Two reviewers (B.J.C. and M.S.M.) independently screened the title and abstract of articles against the inclusion criteria. Full reports were obtained for all studies that were unclear or appeared to meet the inclusion criteria. Full text articles were then independently assessed against the inclusion criteria by each reviewer. Conflicts that arose were resolved through discussion until a consensus was met. When consensus could not be reached, a third reviewer (M.D.M) participated.

### 2.3. Data Extraction and Quality Assessment

The two reviewers independently extracted and recorded data in the Excel Software, including the author and year of publication, population characteristics (sample size, male to female ratio, and age), study design, vitamin C intervention (supplementation/dietary intake, dosage, duration, baseline vitamin C intake), vitamin C levels (serum/plasma sample, baseline and post-intervention vitamin C levels, and the significance of findings), and the collection method. Missing data was noted when contact with the authors was unsuccessful. Analysis was performed using Excel and was discussed narratively.

Bias was assessed independently by two reviewers using the Cochrane Risk of Bias Tool (RoB.2). This tool utilized signalling questions across 5 different domains, including potential bias arising from the randomization process, deviations from intended interventions, missing outcome data, measurement of the outcome, and selection of the reported result. For Domain 2, the effect of assignment to intervention was assessed. The assessment classified studies as ‘low risk’, having ‘some concerns’, or ‘high risk’ for each of the 5 domains and provided a calculated overall bias rating derived with the use of an algorithm provided by the creators of the tool.

## 3. Results

### 3.1. Search Results

The process of selecting appropriate studies is summarized in Figure 1. The search strategy identified 1119 articles; additionally, 17 articles were further identified by searching through the reference list of relevant reviews. After duplicates were removed, 857 articles were screened by title and abstract for their eligibility. From this, 134 articles were assessed by their full text, with 122 articles excluded due to incorrect study design, patient population, intervention, outcome measures, sample size ≤10, or no full text available. Consequently, 11 articles were included in this review, two of which were identified via hand searching.

### 3.2. Study Characteristics

#### 3.2.1. Study Design and Participants

A summary of the study characteristics in the CVD population and healthy population can be found in Table 3 and Table 4, respectively. Of the articles included, six studies recruited CVD patients [23,24,25,26,27,28] and five studies recruited healthy participants [29,30,31,32,33]. Of the CVD studies, CAD was the most common form of CVD studied [23,24,25,27,28]. The sample size ranged from 20–87 participants [24,33], with an average of 31 participants per study. Only males were recruited in two of the 11 studies [23,32], with males being more commonly recruited than females at an average of 73% of the sample population. The age of participants ranged from 23.5–67 years [30,31], with one study specifically looking at older adults over the age of 65 [31]. Of the studies included, six studies reported a double-blind design [23,25,26,27,30,32]; one study was single-blinded, where the participants were aware of the dietary intervention [29]; three studies did not report a blinding format [24,28,33]; while one study reported the single blinding of investigators for dietary intervention and double blinding for supplemental intervention [31]. There were three studies without a placebo intervention in the control group [28,31,33], and one study with a placebo that was not identical to the intervention [29]. Four studies included a cross-over design [26,29,30,32], while the rest of the studies had a parallel design. The oldest study included was published in 1980, with the most recent being published in 2012 [23,29].

#### 3.2.2. Vitamin C Supplementation and Measurement

Most studies utilized oral supplementation as the intervention [23,24,25,27,28,30,32], with only four studies using dietary interventions [26,29,31,33]. Of these studies, two increased fruit and vegetable consumption [31,33] and two utilized drinks, including fruit juice and fruit and vegetable puree beverages (FVPB) [26,29]. The dosage ranged from 250–4500 mg for supplements [25,32], with the lowest vitamin C intervention being 84.5 mg in the FVPB [29]. The dosage range was 84.5–2000 mg per day for the healthy populations, whereas the range of 210–4500 mg was identified for the CVD population. The median daily dose of vitamin C for all studies was 1000 mg. Only two studies acutely administered vitamin C [27,30]. Conversely, the longest supplementation period was 24 weeks [25], with an average duration of 7 weeks across all studies. The duration of vitamin C supplementation was more homogenous in the healthy populations, with three out of six studies administering vitamin C for 6 weeks [29,31,32], and with a range of 1 day to 8 weeks [30,33]. The duration of vitamin C administration had greater variation in the CVD population, ranging from 10 days to 24 weeks [23,24]. Serum vitamin C was measured in three studies [23,24,28], with the remaining studies analyzing plasma concentrations [25,26,27,29,30,31,32,33]. Two studies obtained blood samples 2 h post-intervention [27,30], four after fasting [23,25,28,33], and one after a light meal [26]. Additionally, the fasting state was not described for four studies [24,29,31,32].

### 3.3. Risk of Bias and Study Quality

Based on the Cochrane Risk of Bias Tool 2.0, eight of the studies presented some concerns, and three were judged to be having a high risk of bias, as seen in Figure 2. For the domain pertaining to the randomization process, while all studies mentioned that they incorporated randomization, only three out of eleven studies provided the specific method of randomization utilized [26,29,33]. Two studies showed differences between the control and intervention groups at baseline [27,33], resulting in some concerns within the randomization process. Concerns related to deviation from intended interventions were low as most studies were double-blinded. The forms provided for the placebo and the intervention via vitamin C administration were identical for all but two studies, reducing the chances of differentiation between intervention and control. For all the studies, measurement of the outcome of interest was deemed low-risk as an objective blood test was utilized to determine results. For domain 5, the lack of an available registry in the form of a protocol paper or a statistical analysis plan that pre-specified outcomes of trials was a cause of concern for bias for all studies. Two studies that utilized post hoc analysis and one study that omitted results from a time point were deemed high risk for the selection of reported result. The overall high ratings of bias arose mainly from domain 5 pertaining to the selection of the reported results and lack of documentation of pre-specified outcomes [23].

### 3.4. Change in Vitamin C Levels in Response to Increased Intake

All six studies involving the CVD population reported a significant increase in vitamin C levels in the intervention group (*p* < 0.05) [23,24,25,26,27,28]. Two of the five studies of the healthy population also reported a significant increase in vitamin C levels in response to the intervention (*p* < 0.05) [31,32]. Additionally, two studies of the healthy population where the *p*-value was not reported saw a 216% and 73% increase in plasma vitamin C [30,33]. One study of a healthy population saw a non-significant increase in plasma vitamin C in the intervention group [29]. This study utilized fruit and vegetable puree drinks, with the lowest intervention concentration of vitamin C at 84.5 mg per day [29]. Furthermore, participants had the highest baseline vitamin C levels for all studies (Table 2) [29].

Baseline vitamin C dietary intake was recorded in studies by Bostom et al. (CVD population) [25], George et al. [29], and Zino et al. [33], with the intervention group consuming 95.8, 114 ± 53, and 85 mg per day, respectively.

The mean baseline vitamin C levels for all groups were within the reference range, except for a control group in one CVD study that had plasma vitamin C levels below the reference range [24]. In the healthy population, baseline and post-intervention vitamin C levels in the intervention group ranged from 32.92 ± 13.98 to 89 ± 32 [29,32] and 57.92 ± 22.14 to 135 ± 27 μmol/L, respectively [1,2]. The percentage change ranged from 9% to 216% [29,30]. In the CVD population, baseline and post-intervention vitamin C levels in the intervention group ranged from 28.39 to 57.39 [23,25] and 45.46 ± 1.7 to 116 ± 34 μmol/L [24,27], respectively. The percentage change ranged from 22% to 183% [23,27] across the CVD population studies. Conversely, the control groups saw no significant change in either population.

Notably, significant increases in plasma vitamin C concentration were observed in three of the four dietary intervention studies. Furthermore, the study by Singh et al. [31] included a dietary modification group and a supplementation group. These interventions resulted in a significant increase in plasma vitamin C for both groups, with a 61% and a 63% increase for the dietary and supplementation groups, respectively (Table 2).

## 4. Discussion

From the available literature, this is the first known review to look at the effect of vitamin C intake on the levels of vitamin C in populations with or without CVD. Furthermore, there were no papers identified that compared healthy and CVD patients in order to see how the intake of vitamin C affected vitamin C levels within one study. Therefore, individual studies looking at the effect of vitamin C supplementation within healthy populations and CVD populations were included in this review.

### 4.1. Effect of Vitamin C Supplementation on Post-Intervention Vitamin C Levels

From the results presented, all studies involving the CVD population saw a significant increase in vitamin C levels after intervention. Additionally, all but one study of the healthy population saw a significant increase in vitamin C levels after supplementation. Studies of the CVD population on average had lower baseline ascorbate levels compared to the healthy population, which is consistent with the relevant literature [21,22]. Additionally, average post-intervention ascorbate levels were lower for the CVD population as compared to the healthy population, despite a larger average increase in ascorbate levels. Despite these discrepancies, both population’s baseline and post-intervention vitamin C levels were within the reference ranges for all studies, with the exception of one study of the CVD population [27]. Therefore, the clinical significance of this intervention remains unclear. Studies with patients having lower than normal baseline levels of vitamin C are needed in order to explore clinical significance further.

One study of the healthy population showed no significant increase in plasma vitamin C after the intervention. In this study, it can be seen that the baseline plasma vitamin C levels exceeded the expected maximum plasma levels denoted by pharmacokinetics [13,34]. Plasma vitamin C responds to vitamin C intake in a sigmoid shaped curve, reaching a maximum concentration of around 70–80 μmol/L [13,34]. At this point, plasma vitamin C plateaus due to increased excretion and decreased absorption of vitamin C [13,34]. This homeostatic saturation usually occurs at intakes of 200–400 mg [13,34]. Due to the already high baseline plasma vitamin C levels in this study, it is likely that there was no significant increase in plasma vitamin C in response to the intervention due to near-homeostatic saturation already being reached. This raises an important issue observed in vitamin C supplementation studies, which was explored by Michels and Frei [35]. Studies utilizing subjects with plasma vitamin C levels that are almost or already saturated are unlikely to see additional effects of supplementation on health outcomes [35]. Therefore, Michels and Frei [35] suggested recruiting patients with low plasma vitamin C levels in order to allow maximum opportunity for the intervention to change vitamin C levels. This is therefore an important consideration in the design of intervention studies assessing the impact of vitamin C on CVD outcomes.

### 4.2. Effect of Vitamin C Dosages on Post-Intervention Vitamin C Levels

By making comparisons between post-intervention percentage increases in vitamin C levels in the healthy and CVD populations with similar dosages, and the duration of vitamin C administrations, it can be seen that a greater dose of vitamin C is required in the CVD population to prompt an increase in serum vitamin C levels as compared to the healthy populations.

For instance, while a single vitamin C dose of 2000 mg prompted a 216% increase in serum vitamin C levels in the healthy population [29], a 2500 mg single dose prompted a 183% [27] increase in serum vitamin C in the CVD population. In the CVD populations, 2000 mg doses over 20 days and 4 weeks led to a 96% [23] and a 100% [28] increase in serum vitamin C levels, respectively. Similarly, while 200 mg of administered vitamin C increased the serum vitamin C levels by 100% [32] in the low baseline-plasma healthy population, the same dose administered for four weeks increased serum vitamin C levels by only 28% in the CVD population [26]. From the available data, this suggests that in order to obtain similar increases in vitamin C levels to those of the healthy population, the CVD populations may be required to be administered greater amounts of vitamin C. It can be hypothesized that a greater dosage of vitamin supplementation is required to overcome the greater oxidative stress in CVD populations as compared to healthy populations [36].

### 4.3. Adequate or Optimal Intake Vitamin C Intake

The baseline levels of vitamin C in the CVD population were observed to be lower than those of the healthy population, potentially due to differences in baseline vitamin C intake or the oxidative stress associated with CVD [13]. Despite this, the adequate vitamin C levels in the CVD population could indicate that the habitual intake of participants was enough to overcome the oxidative demands of CVD. Only three studies reported baseline vitamin C dietary intake, one of which was reported in the CVD population [25,29,33]. The reported intake of vitamin C was above the estimated average requirement (EAR) and recommended dietary intake (RDI) for vitamin C in Australia at 30 and 45 mg/day, respectively [37]. The EAR is estimated to provide adequate amounts of vitamin C for smokers, who display an increased turnover of vitamin C, such as that described in CVD populations [13,37]. Therefore, it is possible that meeting or exceeding these recommendations could maintain adequate plasma vitamin C concentrations in the CVD population. However, further research is required to determine an optimum dose of vitamin C needed to improve CVD outcomes as current data shows that the magnitude of increase in serum levels of vitamin C are different in healthy and CVD populations. Furthermore, only two studies included in this review involved population groups >65 years of age. The data available from this review is not externally valid for the elderly population. It is known that incidence and prevalence of PAD and aortic disease rises with increasing age and is more advanced in the elderly population [38,39]. Therefore, further studies of elderly populations need to be conducted.

A minority of the studies included utilized dietary intervention, with all but one of these interventions resulting in a significant increase in vitamin C levels. Furthermore, one study [31] looked at diet and supplementation compared to the control. The results highlighted that despite the supplementation dose being larger than the dietary vitamin C intake, plasma vitamin C rose to the same level in both groups. This finding could be attributed to homeostatic saturation being achieved at lower intakes, therefore resulting in the increased excretion and decreased absorption of vitamin C from the high-dose supplement [34]. This highlights that dietary interventions can be a useful tool to increase vitamin C intake and the corresponding levels, and that excessive high-dose supplementation may not provide additional benefits. Vitamin C, at high levels, can act as a pro-oxidant through the reduction of ferric ions, which may be more available in atherosclerosis due to tissue disruption, which can then generate ROS [5]. Additionally, adverse gastrointestinal effects may be experienced with acute, high doses of vitamin C [37]. This reflects the importance of avoiding over supplementation. Therefore, further research to determine an optimal dose is indicated.

### 4.4. Quality Assessment

The quality assessment, as depicted in Figure 2, highlighted that the studies presented had some concerns or a high risk of overall bias. The use of an objective measurement to determine outcome provides confidence to the results. Conversely, the lack of registered pre-specified outcomes of the studies presents a risk of bias. This is because the selection of a reported outcome could be presented based on favorable findings and must therefore be used with caution.

### 4.5. Strengths and Limitations

This review was able to identify gaps in the literature, such as the lack of studies on the effectiveness of vitamin C supplementation increasing levels of vitamin C in the CVD population as compared to those in healthy controls. However, the lack of articles comparing the levels of vitamin C in the healthy and CVD population in response to the same dosage and duration of vitamin C intake also became a major limitation of this review, thus making it difficult to draw conclusions from the data. Additionally, the high risk of bias in multiple studies, and the different forms of vitamin C doses and durations used between the studies restrict the ability to determine an optimal regimen for increasing plasma concentrations in each population. Furthermore, the findings of this review may not be applicable to all subgroups of CVD due to the lack of studies of the PAD population, which tended to be the most advanced form of CVD, carrying a greater burden of oxidative stress [4,40]. Another possible limitation pertains to the likelihood of naturally occurring inter-individual differences in vitamin C levels, which may not have been detected due to the small sample size of the studies included.

This review highlights the lack of literature that compares the response of plasma vitamin C to vitamin C intake in the healthy and CVD populations. It has also illustrated the importance of participants in vitamin C intervention studies having low to moderate baseline vitamin C levels in order to allow for maximum opportunity for the intervention to increase plasma concentration. Consequently, studies designed in this way will have a greater opportunity to observe the effect of vitamin C intervention on CVD outcomes. Furthermore, results from this study also show that greater doses of vitamin C supplementation are required to obtain similar increases in plasma vitamin C levels in CVD populations as compared to healthy populations. Therefore, it is recommended that future studies investigate the optimal vitamin C regimen for replenishing low baseline plasma vitamin C in patients with CVD as compared to the healthy population, including a PAD subgroup. This is of particular interest as pharmacokinetics already established in the healthy population may not be applicable to populations with diseases due to factors such as differences in metabolism [35]. Findings from these future studies could then be used to inform the design of vitamin C intervention studies in the CVD population, and potentially contribute to vitamin C repletion recommendations for the CVD population. The review also highlights the lack of similar research conducted in elderly populations who tend to have more advanced CVD.

## 5. Conclusions

In conclusion, the increased oxidative stress associated with the CVD disease process raised the question of whether vitamin C intake used in experimental studies was sufficient to increase levels of vitamin C in the CVD population. This novel review highlights that levels of vitamin C increased significantly for nearly all groups with or without CVD. Due to the limitations of poor study quality and differing intervention regimens, the results presented must be used with caution. However, this review has importantly highlighted the need for future research to investigate how plasma vitamin C responds to vitamin C intake in CVD populations with low to moderate baseline vitamin C levels. This information could then be used to inform the design of future intervention studies or vitamin C repletion guidelines. Further similar research in older populations is also warranted.

## Figures and Tables

**Figure 1 nutrients-13-02330-f001:**
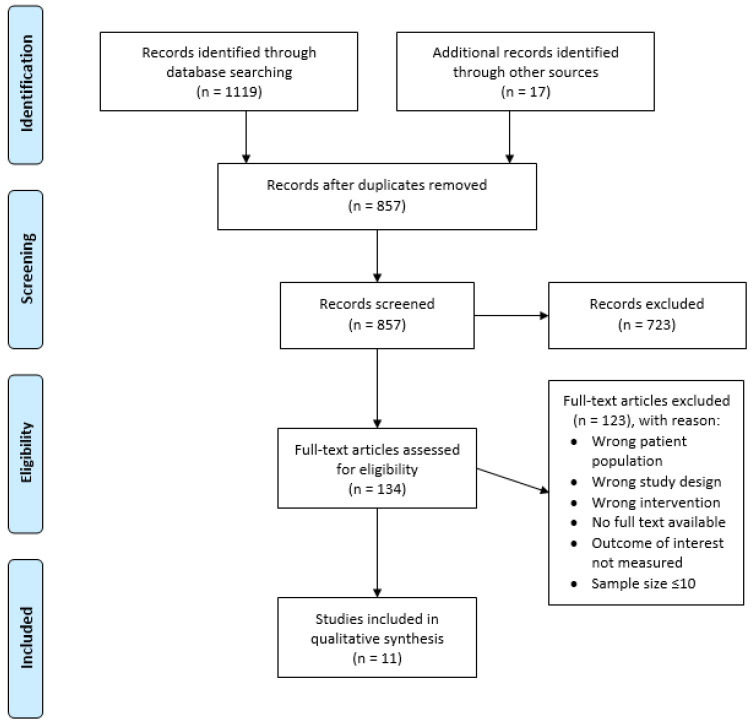
PRISMA flow diagram outlining the study selection process.

**Figure 2 nutrients-13-02330-f002:**
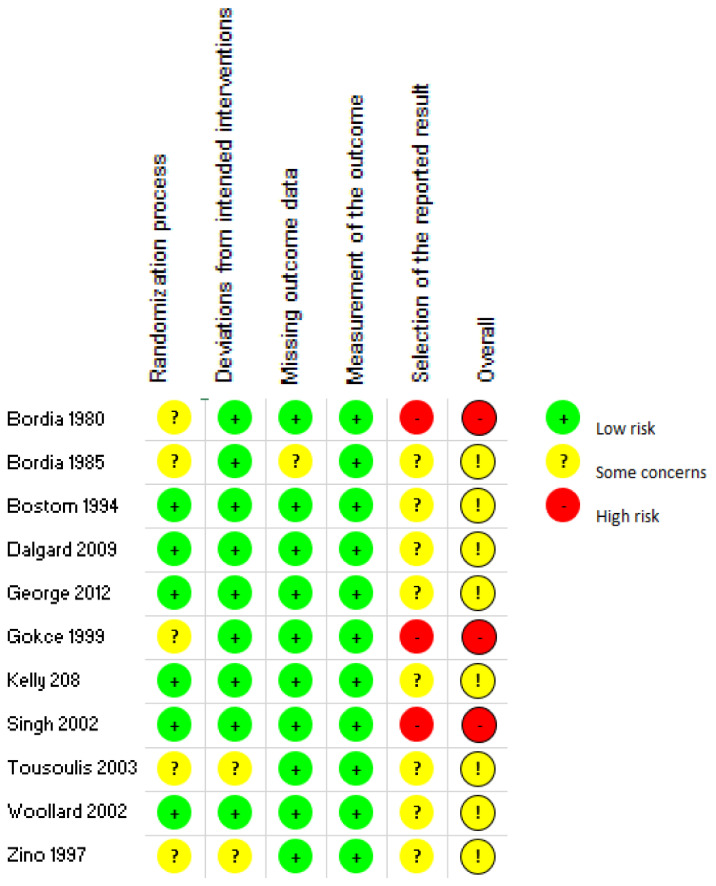
Summary of risk of bias results.

**Table 1 nutrients-13-02330-t001:** Full details of the search terms used across databases.

Concept ^a^	Search Terms
Concept 1	(“Vitamin C” OR Ascorb OR Antioxidant).ti,ab
Concept 2	Intake OR Consumption OR “Dietary intake” OR Supplement OR Supplementation OR “Nutritional supplement”
Concept 3	“Plasma vitamin C” OR “Plasma ascorb” OR “Serum vitamin C” OR “Serum ascorb” OR “Vitamin C level” OR “Ascorb level” OR “Vitamin C concentration” OR “Ascorb concentration”
Concept 4	(“Cardiovascular disease” OR CVD OR “Coronary heart disease” OR CHD OR “Ischaemic heart disease” OR “Ischemic heart disease” OR IHD OR “Coronary artery disease” OR CAD OR “Peripheral artery disease” OR PAD OR “Peripheral vascular disease” OR PVD).ti,ab

^a^ Concept 1, 2, 3, and 4 joined with ‘AND’.

**Table 2 nutrients-13-02330-t002:** Inclusion and Exclusion criteria.

Inclusion Criteria	Exclusion Criteria
Studies that are randomized controlled trials	Studies including participants at risk of CVD, including otherwise healthy smokers and hypertensive patients
Studies with a sample size greater than 10 participants	Studies that administer mixed nutrient supplementation where no group received vitamin C alone
Studies including participants aged 18 or above	Studies incorporating dual treatments such as exercise and supplementation
Studies including participants who are either healthy or have established CVD, defined as a clinically manifested disease of the blood vessels including PAD or CAD	
Studies that orally administer a single nutrient intervention of vitamin C through supplementation or dietary intervention resulting in vitamin C intake ≥45 mg/day	
Studies that record percentage change in vitamin C levels or record before and after intervention levels of vitamin C	

**Table 3 nutrients-13-02330-t003:** Data summary table of studies investigating vitamin C intake and vitamin C levels in CVD populations.

Author and Year	Study Design	Disease Status	Sample Size Mean (%male)	Age Mean ± SD	Intervention Type and Dosage	Duration	Control	Baseline Vitamin C Levels ± SD (μmol/L)	Post-Intervention Vitamin C Levels ± SD (μmol/L) (*p*-Value) d	Percentage Change in Vitamin C Levels (%) c
Bordia 1980 [23]	P1: DB, RCT P2: RCT	P1: CAD P2: Acute MI	P1: 40 (100) P2: 40 (NR)	Intervention P1: G2: 47.2 G3: 50.8 Control P1: 52.7 P2: NR	Supplement P1: G2—500 mg, BD G3—1000 mg, BD P2: 1000 BD	P1: 6 months P2: 20 days	P1 and 2: Placebo vitamin C capsules	P1: NR Intervention P2: 28.39a Control P2: NR	Intervention and control for P1 and P2: NR	Intervention P1: G2: 22 (*p* < 0.05) d G3: 96 (*p* < 0.001) d Intervention P2: 94 (*p* < 0.001) d Control P1 and P2: NR
Bordia and Verma 1985 [24] (Part 2)	RCT	CAD	20 (NR)	NR	Supplement 1000 mg, every 8 h	10 days	Placebo vitamin C	Intervention: 28.98 ± 2.84b Control: 22.73 ± 5.68b	Intervention: 45.46 ± 1.7b (*p* < 0.001) Control: 24.43 ± 1.7b (NR)	Intervention: 57 Control: 7
Bostom et al., 1994 [25]	DB, RCT	CAD	44 (NR)	NR	Supplement 4500 mg	12 weeks	Matching placebo	Intervention: 57.39b Control: 57.39b	Intervention: 126.71b (*p* = 0.0001) Control: 65.91b (NR)	Intervention: 121 Control: 15
Dalgard et al. 2009 [26]	BR, DB, CO	PAD	48 (46)	Average: NR Intervention:JP: 57.4 ± 6.2 Control: RP: 60.8 ± 5.8	Dietary orange and blackcurrant juice ~210 mg	4 weeks	Sugar containing reference beverage (0 mg vit c)	Intervention: 46.4 Control: 31.5	Intervention: 59.4 (*p* < 0.005) Control: 28.6 (NR)	Intervention: 28 Control: −9
Gokce et al., 1999 [27]	DB, RCT	CAD	46 (91)	NR Intervention: 54 ± 9 Control: 56 ± 12	Supplement 2000 mg single dose + 500 mg daily	1 month	Matching placebo	Intervention: 41 ± 13 Control: 43 ± 19	Intervention: Acute: 116 ± 34 (*p* < 0.05) 30 days: 95 ± 36 (*p* < 0.05) Control: Acute: 43 ± 20 (NR)30 days: 38 ± 20 (NR)	Intervention: Acute: 183 30 days: 132 Control: Acute: 0 30 days: −12
Tousoulis et al., 2003 [28]	RCT	T2DM and CAD	39 (87)	Average: NR Intervention: 63.3 ± 2.7 Control: 67.4 ± 2.1	Supplement 2000 mg	4 weeks	No antioxidant treatment	Intervention: 39.9 ± 2.0 Control: 39.5 ± 2.1	Intervention: 79.9 ± 3.6 (*p* = 0.001) Control: 40.0 ± 2.2 (*p* = 0.755)	Intervention: 100 Control: 1

DB, double-blind; SB, single-blind; CO, cross over; BR, block-randomization; RCT, randomized control trial; MI, myocardial infarction; CAD, coronary artery disease; PAD, peripheral artery disease; T2DM, type 2 diabetes mellitus; BD, twice daily; JP, Juice + Placebo vit E; RP, Reference Beverage + Placebo vit E; NR, not reported; NS, not significant; SD, standard deviation. a Converted from mg/100 mL to μmol/L. b Converted from mg/dL to μmol/L. c Calculated using Excel for all studies except [23]. d *p*-value obtained from studies.

**Table 4 nutrients-13-02330-t004:** Data summary table of studies investigating vitamin C intake and vitamin C levels in the healthy population.

Author and Year	Study Design	Health Status	Sample Size Mean (%male)	Age Mean ± SD	Intervention Type and Dosage	Duration	Control	Baseline Vitamin C Levels ± SD (μmol/L)	Post-Intervention Vitamin C Levels ± SD (μmol/L) (*p*-Value) d	Percentage Change in Vitamin C Levels (%) c
George et al., 2012 [29]	SB, CO, RCT	Healthy	39 (38)	Average: 45 Intervention: NR Control: NR	Dietary-FV puree drinks 84.5 mg	6 weeks	Diluted fruit-flavored cordial 2 × 50 mL (not identical to intervention)	Intervention: 89 ± 32 Control: 91 ± 37	Intervention: 97 ± 44 (NS) Control: 81 ± 36 (NR)	Intervention: 9 Control: −11
Kelly et al., 2008 [30]	DB, RCT, CO	Healthy	26 (75)	Average: 23.5 ± 1.4 Intervention and Control: NR	Supplement 2000 mg	Single dose	Water flavored with lemon juice.	Intervention: 38 ± 18 Control: 40	Intervention: 120 ± 26 (NR) Control: 40 (NR)	Intervention: 216 Control: 0
Singh et al., 2002 [31]	RCT, DB and SB (diet)	Healthy	56 (46)	Average: 67± 1 Intervention: Diet: 66 Sup: 66 Control: 69	Dietary (Diet) extra 3 FV portions/day. Extra ~210 mg Supplement 1000 mg	6 weeks	Diet: no control. Sup: matching placebo tablets	Intervention: Diet: 84 ± 5 Sup: 83 ± 4 Control: 83 ± 5	Intervention: Diet: 135 ± 27 (*p* < 0.05) Sup: 135 ± 8 (*p* < 0.05) Control: 88 ± 9 (NR)	Intervention: Diet: 61 Sup: 63 Control: 6
Woollard et al., 2002 [32]	DB, CO, RCT	Healthy	40 (100)	Average: 30 Intervention and Control: NR	Supplement 250 mg	6 weeks	Vitamin C placebo	Average: 50.4 ± 3.4 Intervention: LOC: 32.92 ± 1 3.98 HIC: 67.61 ± 6.32	Intervention: LOC: Reported 2-fold increase = ~65.84 (*p* < 0.0001) HIC: 95.32 ± 25.92 (*p* < 0.0005)	Intervention: LOC: 100 HIC: 41
Zino et al., 1997 [33]	RCT	Healthy	87 (71)	Average: NR Intervention: 28.6 ± 7.9 Control: 31.9 ± 11.4	Dietary—8 serve FV/day. Extra 172 mg	8 weeks	No dietary intervention	Intervention: 33.50 ± 21 Control: 25.55 ± 21.58	Intervention: 57.92 ± 22.14 Control: 25.55 ± 20.44	Intervention: 73 (NR) Control: 0 (NR)

DB, double-blind; SB, single-blind; CO, cross over; BR, block-randomization; RCT, randomized control trial; FV, fruit and vegetable; LOC, low baseline plasma vitamin C group; HIC, high baseline plasma vitamin C group; NR, not reported; NS, not significant; SD, standard deviation. c Calculated using Excel for all studies. d *p*-value obtained from studies.

## Data Availability

Data is contained within the article or is available from the included studies that have been referenced throughout.

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
