# Peer review of "Effect of Dietary or Supplemental Vitamin C Intake on Vitamin C Levels in Patients with and without Cardiovascular Disease: A Systematic Review"

_nutrients, 2021, doi:10.3390/nu13072330_

Round 1

Reviewer 1 Report

The review deals with an interesting medical problem and is interestingly elaborated, but not very original. We can find similar research, for example, in the article: Vitamin C supplementation for the primary prevention of cardiovascular disease. Al-Khudairy L, Flowers N, Wheelhouse R, Ghannam O, Hartley L, Stranges S, Rees K.Cochrane Database Syst Rev. 2017 Mar 16;3(3):CD011114. doi: 10.1002/14651858.CD011114.pub2.

In Introduction:

The role of vitamin E and its interaction with vitamin C, as well as the influence of these vitamins on the level of lipid peroxidation, are missing. There is no literature review on supplementation with vitamins E and C in patients with CVD. Perhaps it is also worth mentioning the pro-oxidative effect of vitamin C administered in large doses.

Reference ranges of serum Vitamin C levels are 0.4-1.5 mg / dL or 23-85 μmol / L is not representative of the entire population. For Europeans it is 50-75 μmol / L, other sources provide other reference ranges.

In Discussion:

Very general conclusions, theses. Using supplementation, you can not explain the differences observed in the concentrations of plasma vitamin C in patients with CVD, oxidative stress, because practically not studied this parameter. Perhaps the main problem of interpretation, it is very small numbers of cases evaluated in the works selected for review and naturally occurring inter-individual differences in the concentrations of vitamin C.

Reviewer 2 Report

This review deals with a very interesting topic.

Overall is well organised and with a great reading flux.

I only two suggestions before the final publication of the paper:

  • the authors used google translator for the non-english papers. Be careful with this as google translator not always have a satisfactory accuracy (mainly for languages with different alphabet such as Chinese) and can completely change the original meaning. With that said, I encourage the authors to state in the methods whether any of the non-english papers were utilised.
  • I think the review would greatly benefit with the insertion of a schematic illustration of the main conclusions.

Round 2

Reviewer 1 Report

I'm satisfied with changes that have been made to the manuscript.